# Emerging cooperativity between Oct4 and Sox2 governs the pluripotency network in early mouse embryos

Yanlin Hou[1,2†], Zhengwen Nie[2], Qi Jiang[2], Sergiy Velychko[3], Sandra Heising[1], Ivan Bedzhov[4], Guangming Wu[2*], Kenjiro Adachi[1*‡], Hans R Scholer[1*]

[1]Cell and Developmental Biology Group, Max Planck Institute for Molecular Biomedicine, Münster, Germany; [2]Guangzhou National Laboratory, Guangzhou International Bio Island, Guangzhou, China; [3]Department of Genetics, Harvard Medical School, Boston, United States; [4]Embryonic Self-Organization Research Group, Max Planck Institute for Molecular Biomedicine, Münster, Germany

*For correspondence:
wu_guangming@gzlab.ac.cn
(GW);
kenjiro.adachi@astellas.com
(KA);
h.schoeler@mpi-muenster.mpg.
de (HRS)

Present address: [†]Smilow Center for Translational Research, University of Pennsylvania, Philadelphia, United States; [‡]Astellas Pharma Inc, Tsukuba, Japan

**Competing interest:** The authors declare that no competing interests exist.

## eLife Assessment

This study presents a **valuable** finding on how the interplay between transcription factors SOX2 and OCT4 establishes the pluripotency network in early mouse embryos. The evidence supporting the claims of the authors is **solid**, although inclusion of additional omics data would further strengthen the study. The work will be of interest to biologists working on embryonic development and gene regulation.

**Abstract** During the first lineage segregation, mammalian embryos generate the inner cell mass (ICM) and trophectoderm (TE). ICM gives rise to the epiblast (EPI) that forms all cell types of the body, an ability referred to as pluripotency. The molecular mechanisms that induce pluripotency in embryos remain incompletely elucidated. Using knockout (KO) mouse models in conjunction with low-input ATAC-seq and RNA-seq, we found that Oct4 and Sox2 gradually come into play in the early ICM, coinciding with the initiation of Sox2 expression. Oct4 and Sox2 activate the pluripotency-related genes through the putative OCT-SOX enhancers in the early ICM. Furthermore, we observed a substantial reorganization of chromatin landscape and transcriptome from the morula to the early ICM stages, which was partially driven by Oct4 and Sox2, highlighting their pivotal role in promoting the developmental trajectory toward the ICM. Our study provides new insights into the establishment of the pluripotency network in mouse preimplantation embryos.

## Introduction

Following fertilization, mouse embryos undergo a series of cell divisions and differentiation in the first few days. Zygotes and 2-cell blastomeres are totipotent and can form both embryonic and extra-embryonic supporting tissues. After several rounds of divisions, embryos undergo compaction at the morula stage and specifies the trophectoderm (TE) and inner cell mass (ICM). The ICM further segregates into the primitive endoderm (PE) and the epiblast (EPI) in the subsequent blastocyst stage. All cell types of the fetus can develop from the EPI, a feature known as pluripotency. During the establishment of pluripotency, dramatic molecular changes occur such as shifts in epigenetic modifications, chromatin accessibility, and transcriptome (*Deng et al., 2014*; *Guo et al., 2010*; *Wang et al., 2018*; *Wu et al., 2016*; *Zhang et al., 2016*; *Zhang et al., 2018*). The precise mechanisms involved in establishing the transient pluripotent state during development remain elusive.

Oct4 (encoded by *Pou5f1*) and Sox2 are prominent transcription factors (TFs) that regulate early embryonic development. In mice, Oct4 is expressed in all the cells of the compacted morulae, while Sox2 is first expressed in the inside cell and identified as one of the earliest markers to distinguish the inner from the outer cells (*Guo et al., 2010*; *Palmieri et al., 1994*; *White et al., 2016*). Both factors are confined to the pluripotent EPI at the late blastocyst stage (*Avilion et al., 2003*; *Palmieri et al., 1994*). *Pou5f1* and *Sox2* knockout (KO) leads to embryonic lethality around embryonic day (E) 4.5 and E6.0, respectively (*Avilion et al., 2003*; *Nichols et al., 1998*). Although *Pou5f1-* or *Sox2*-KO embryos develop apparently normal EPI, they fail to give rise to embryonic stem cells (ESCs) (*Avilion et al., 2003*; *Nichols et al., 1998*) and *Pou5f1*-KO embryos cannot contribute to embryonic tissues in chimera assays (*Wu et al., 2013*), suggesting that Oct4 and Sox2 are required for the EPI to acquire or maintain pluripotency. Surprisingly, *Pou5f1-* and *Sox2-KO* mouse embryos still express many pluripotency markers, such as Nanog, *Klf4,* and *Zfp42* (*Rex1*) (*Avilion et al., 2003*; *Frum et al., 2013*; *Le Bin et al., 2014*; *Stirparo et al., 2021*; *Wicklow et al., 2014*; *Wu et al., 2013*). The controversy between the loss of pluripotency and the maintenance of the pluripotency-related genes in *Pou5f1-* and *Sox2-KO* EPI prompted us to investigate the role of Oct4 and Sox2 in early mouse embryos.

Recent studies suggested that Oct4 and Sox2 may function before the blastocyst stage. The long-lived presence of exogenous Oct4 and Sox2 proteins on DNA was observed in some of the 4-cell blastomeres, that tended toward the ICM rather than the TE (*Plachta et al., 2011*; *White et al., 2016*), suggesting that they may play a role in the initial lineage segregation between ICM and TE. Oct4 is also thought to contribute to the dramatic gain of chromatin accessibility at the 8-cell stage (*Lu et al., 2016*). Unlike in mouse embryos, the loss of OCT4 expression in human embryos compromises blastocyst development, leading to the downregulation of *NANOG* in the ICM and *CDX2* in the TE (*Fogarty et al., 2017*). Oct4 controls zygotic genome activation in zebrafish and human, but not in mice (*Gao et al., 2018*; *Leichsenring et al., 2013*). Thus, when and how Oct4 and Sox2 function in the early embryogenesis varies from species to species.

Oct4 and Sox2 play preeminent roles in maintaining the pluripotency-related transcriptional network in ESCs (*Masui et al., 2007*; *Niwa et al., 2000*). However, it remains unclear how these two TFs regulate pluripotency-related genes in embryos. The obvious obstacles are the heterogeneity and the scarcity of the preimplantation embryos. Unlike ESCs, which comprise a homogenous cell population, individual cells within embryos gradually differentiate into three cell types, EPI, PE, and TE. This inherent cellular heterogeneity within embryos, coupled with the temporal variability in development across different embryos, poses a significant challenge when investigating the molecular processes using conventional molecular techniques. For instance, using the entire blastocyst or ICM for qPCR and bulk RNA-seq analysis would inevitably lead to a dilution and blur of the molecular processes within EPI (*Frum et al., 2013*; *Wu et al., 2013*). Although single-cell genomics is powerful to dissect cellular heterogeneity, it suffers from high dropouts, which may lead to inaccuracies in genomic profiles and cell identities (*Minow et al., 2023*). Due to the scarcity of embryos, it is unfeasible to compensate for the inaccuracy by increasing the number of cells. For example, single-cell RNA-seq (scRNA-seq) still faces challenges in accurately distinguishing between ICM and TE progenitors in compacted morula, as well as between EPI and PE progenitors in the early ICM (*Deng et al., 2014*; *Li et al., 2023*; *Yanagida et al., 2022*).

MEK inhibitor (MEKi) promotes the ground state of ESCs by suppressing the PE-inducing ERK signaling pathway (*Ying et al., 2008*). In embryos, MEKi has been shown to activate Nanog in ICMs, effectively suppressing the development of PE and directing the entire ICM toward a pluripotent EPI fate (*Nichols et al., 2009*). The naïve pluripotency and readiness for differentiation of MEKi-treated EPI has been confirmed by its contribution to chimaeras with germline transmission. MEKi may thus facilitate the study of molecular events within the EPI cells by reducing the heterogeneity in ICMs.

In this study, we analyzed the transcriptome and global chromatin accessibility in the *Pou5f1-* or *Sox2*-KO mouse embryos using low-input RNA-seq and ATAC-seq. The effects of *Pou5f1*-KO and *Sox2*-KO are relatively small in morulae, but become evident in ICMs at the blastocyst stage. Oct4 and Sox2 mainly activate pluripotency-related genes cooperatively through the putative enhancers containing the composite OCT-SOX motifs. We observed a substantial reorganization of open chromatin regions and transcriptome in the early ICM, which was disturbed in absence of either Oct4 or Sox2. These results indicate the critical roles of Oct4 and Sox2 in establishing the pluripotency network during early mouse development.

# Results

## Generation of maternal-zygotic KO embryos

To investigate the role of Oct4 and Sox2 in the development of preimplantation embryos, we produced four transgenic mouse lines: *Pou5f1*-KO labeled with mKO2 (monomeric kusabira-orange 2, *Pou5f1^mKO2^*), *Sox2*-KO labeled with EGFP (*Sox2^EGFP^*), floxed *Pou5f1* (*Pou5f1^flox^*), and floxed *Sox2* (*Sox2^flox^*) (*Figure 1A and B*; Materials and methods). Although the expression of mKO2 and EGFP is driven by the constitutive mouse PGK promoter, we observed that the expression of these reporters is weak until the 8-cell and blastocyst stages, respectively, when the strong zygotic expression of Oct4 and Sox2 is observed.

*Pou5f1* ^+/mKO2^ and *Sox2* ^+/EGFP^ heterozygous male mice were mated with *Pou5f1* ^flox/flox^; *Zp3-Cre* and *Sox2* ^flox/flox^; *Zp3-Cre* maternal KO female mice, respectively, to generate fluorescently labeled maternal-zygotic KO and unlabeled maternal KO (herein called control, Ctrl) embryos (*Figure 1C and D*). These reporter systems allow prospective identification of KO embryos using fluorescent microscopy (*Figure 1—figure supplement 1*, *Figure 1—figure supplement 1—source data 1*), so we were able to pool embryos with the same genotype. Immunostaining results confirmed that the mKO2+ and EGFP+ embryos lost Oct4 and Sox2 proteins, respectively (*Figure 1E and F*, *Figure 1—source data 2*, *Figure 1—source data 3*).

## Oct4 and Sox2 regulate the chromatin landscape and transcriptome in the ICM

To assess the genome-wide molecular impact of the loss of Oct4 and Sox2, we performed low-input ATAC-seq and RNA-seq using early (E2.75) and late (E3.25) compacted morulae, as well as early (E3.75) and late (E4.5) ICMs (Materials and methods). For ATAC-seq, we pooled embryos based on our reporter systems because single-embryo samples yielded sparse signals and too few peaks (data not shown). We opted to collect single morulae or ICMs for RNA-seq, as this approach enabled us to account for embryo-to-embryo variability and detect transcripts with greater sensitivity compared to scRNA-seq (*Boroviak et al., 2015*). Given that the ICM consists of the progenitors of EPI and PE cells, we treated the embryos with the MEKi (PD0325901) from E2.5 to suppress PE development and specify the entire ICM to the EPI (*Nichols et al., 2009*; *Figure 1—figure supplement 2*, *Figure 1—figure supplement 2—source data 1*).

Principal component analysis (PCA) of the ATAC-seq data reveals a clear clustering of samples based on the stages and genotypes (*Figure 2A*). The *Pou5f1*-KO early and late morulae clustered close to their Ctrl counterparts, while the *Pou5f1*- and *Sox2*-KO early and late ICM samples clustered separately from their Ctrl. In total, 152,393 peaks were identified across different stages and genotypes. Of these, 14,016 (9.2%) and 6637 (4.4%) showed significant decreases, while 11,884 (7.8%) and 3660 (2.4%) exhibited significant increases in *Pou5f1*- and *Sox2*-KO late ICM, respectively (*Figure 2B*; *Supplementary file 4*). The number of decreased peaks exceeded that of increased peaks across all the KO samples. K-means clustering of the 33,520 differential peaks identifies subsets of open chromatin regions that were differently affected by *Pou5f1*- and *Sox2*-KO (*Figure 2C*). Notably, compared to the peaks more reliant on Oct4 than Sox2 (*Figure 2B*, clusters 1 and 11), those highly reliant on both Oct4 and Sox2 (clusters 3, 8, and 14) show greater enrichment of the OCT-SOX motif. The former group tended to be already open in the morula, while the latter group became open in the ICM. Over 96.8% of the significantly changed peaks were identified as putative enhancers located distally to the transcription start sites (TSSs) (*Figure 2—figure supplement 1A*).

To validate the authenticity of the Sox2-regulated downstream ATAC-seq peaks identified in this study, we compared our data to a recent study that did not employ MEKi (*Li et al., 2023*). The trends observed in our *Sox2*-KO late ICMs, including decreased, increased, and unchanged peaks, remained consistent in the absence of MEKi (*Figure 2—figure supplement 1B*). Remarkably, the decreased ATAC-seq peaks were enriched with Sox2 CUT&RUN signals (*Figure 2—figure supplement 1C*). Analysis of published ChIP-seq data in ESCs (*Marson et al., 2008*; *Whyte et al., 2012*) shows that the peaks decreasing in the *Pou5f1*- or *Sox2*-KO late ICMs were frequently bound by Oct4, Sox2, and Nanog, and enriched with the active enhancer mark H3K27ac (*Figure 2—figure supplement 1D*). The Sox2-dependent peaks around the *Sap30* gene are shown as examples (*Figure 2—figure supplement 1E*). These data suggest that the decreased peaks identified in our ATAC-seq could be the potential targets of Oct4 and Sox2 in the ICM.

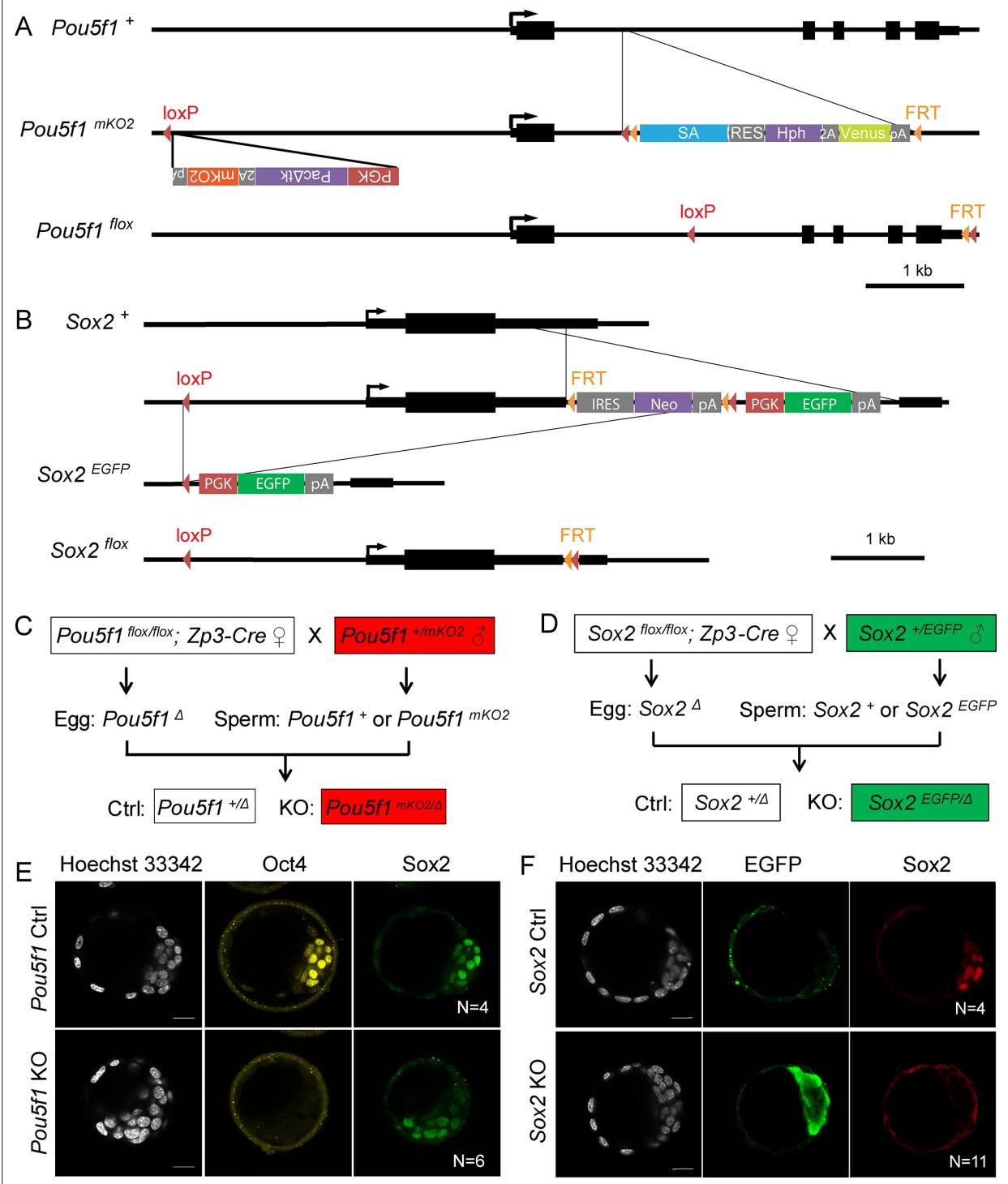

**Figure 1.** Generation of maternal-zygotic knockout (KO) embryos. (**A**) Schemes of mKO2-labeled *Pou5f1*-KO (*Pou5f1^mKO2^*) and *Pou5f1* ^flox^ alleles. (**B**) Schemes of EGFP-labeled *Sox2*-KO (*Sox2^EGFP^*) and *Sox2* ^flox^ alleles. (**C**) Mating strategy for *Pou5f1* Ctrl and KO embryos. (**D**) Mating strategy for *Sox2* Ctrl and KO embryos. (**E**) Immunostaining with embryos separated based on the mKO2 fluorescence. (**F**) Immunostaining with embryos separated based on the EGFP fluorescence. N, the number of embryos. Scalebar, 20 μm.

The online version of this article includes the following source data and figure supplement(s) for figure 1:

**Source data 1.** Imaging data of Pou5f1+ mKO2- embryos in panel E.

**Source data 2.** Immunostaining of the Pou5f1- mKO2+ embryo in panel E.

*Figure 1 continued on next page*

*Figure 1 continued*

**Source data 3.** Imaging data of Sox2+ EGFP- and Sox2- EGFP+ embryos in panel F.

**Figure supplement 1.** Validation of the transgenic embryos.

**Figure supplement 1—source data 1.** Live imaging of embryos.

**Figure supplement 2.** Validation of the effect of MEK inhibitor (MEKi).

**Figure supplement 2—source data 1.** Imaging data of embryos.

To assess the impact of *Pou5f1* and *Sox2* deletion on the transcriptome, we performed RNA-seq on individual morulae or ICMs. Maternal KO embryos (circles in *Figure 2—figure supplement 2A*) clustered together with wildtype embryos (triangles and squares) in the PCA, consistent with previous studies reporting no observable phenotype in maternal KO embryos (*Avilion et al., 2003*; *Frum et al., 2013*; *Kehler et al., 2004*; *Nichols et al., 1998*; *Wicklow et al., 2014*; *Wu et al., 2013*). Therefore, we employed maternal KO as our Ctrl group. In morulae, *Pou5f1*-KO had only a minor effect on the transcriptome (*Figure 2D*). We observed a progressively profound impact of *Pou5f1*-KO on the transcriptome from the morula to the late ICM, mirroring patterns at the chromatin level (*Figure 2A–D*). As Sox2 is only expressed at very low levels until the blastocyst stage anyway, the transcriptome in *Sox2*-KO morulae was not significantly affected. *Pou5f1*- or *Sox2*-KO altered the expression of 2485 (15.8%) and 967 (6.1%) genes in the late ICM, respectively (*Figure 2D*; *Supplementary file 5*). *Pou5f1*-KO had a greater effect than *Sox2*-KO on both the chromatin accessibility and the transcriptome (*Figure 2A–D*), suggesting Oct4 may play a more important role than Sox2. This is consistent with the earlier developmental arrest of *Pou5f1*-KO embryos at E4.5 compared to *Sox2*-KO embryos at E6.0. Down- and upregulated genes identified in our *Pou5f1*- and *Sox2*-KO ICMs show enrichment with genes exhibiting down- and upregulation, respectively, in the studies without MEKi (*Li et al., 2023*; *Stirparo et al., 2021*; *Figure 2—figure supplement 2B–D*), confirming that MEKi did not alter the molecular functions of Oct4 and Sox2 in the ICM.

Next, we explored whether changes in chromatin accessibility affected the transcription of nearby genes, specifically those with TSSs located within 10 kb of the peak centers. Gene set enrichment analysis (GSEA) shows that the genes close to the decreased peak centers were frequently downregulated, whereas those near the increased peak centers were upregulated in the *Pou5f1*- and *Sox2*-KO embryos (*Figure 2E*; *Figure 2—figure supplement 3*). For example, the expression of *Utf1*, a known target of Oct4 and Sox2, as well as the accessibility of its well-characterized OCT-SOX enhancer decreased in the *Pou5f1*- and *Sox2*-KO ICMs (*Nishimoto et al., 1999*; *Figure 2F and G*). On the other hand, the expression of *Fgfr2* and the accessibility of ATAC-seq peaks within its introns increased in the *Pou5f1*- and *Sox2*-KO ICM. In addition, integration of the ATAC-seq and RNA-seq data allowed us to infer previously unknown targets of Oct4 and Sox2, such as *Sap30* and *Uhrf1*, which are essential for somatic cell reprogramming and embryonic development (*Figure 2F and G*; *Cao et al., 2019*; *Li et al., 2017*; *Maenohara et al., 2017*).

Taken together, the data presented so far indicate that Oct4 and Sox2 play a crucial role in shaping the chromatin landscape and transcriptome in the ICM. Consequently, we observed apparently normal development of the *Pou5f1*- and *Sox2*-KO embryos up to the blastocyst stage.

## Oct4 and Sox2 activate the pluripotency network in the ICM

To further explore the role of Oct4 and Sox2, we wanted to find out which genes they might influence in the ICM. As expected, many of the differential ATAC-seq peaks were consistently changed in *Pou5f1*- and *Sox2*-KO late ICMs (*Figure 3A*, *Supplementary file 4*). However, there are a number of peaks exclusively changed in either *Pou5f1*- or *Sox2*-KO ICMs. The genes around the consistently decreased peaks were enriched for terms related to pluripotency and preimplantation embryonic development, such as cellular response to LIF signaling and stem cell population maintenance (*Figure 3B*, upper panel). The consistently elevated peaks preferentially located near the genes related to extraembryonic lineages and organogenesis, such as embryonic placenta, extraembryonic trophoblast, muscle, and neural tube (*Figure 3B*, lower panel). Interestingly, the genes near the peaks which decreased only in the *Pou5f1*-KO but not in the *Sox2*-KO ICM were enriched with terms of LIF signaling and blastocyst formation (*Figure 3—figure supplement 1*).

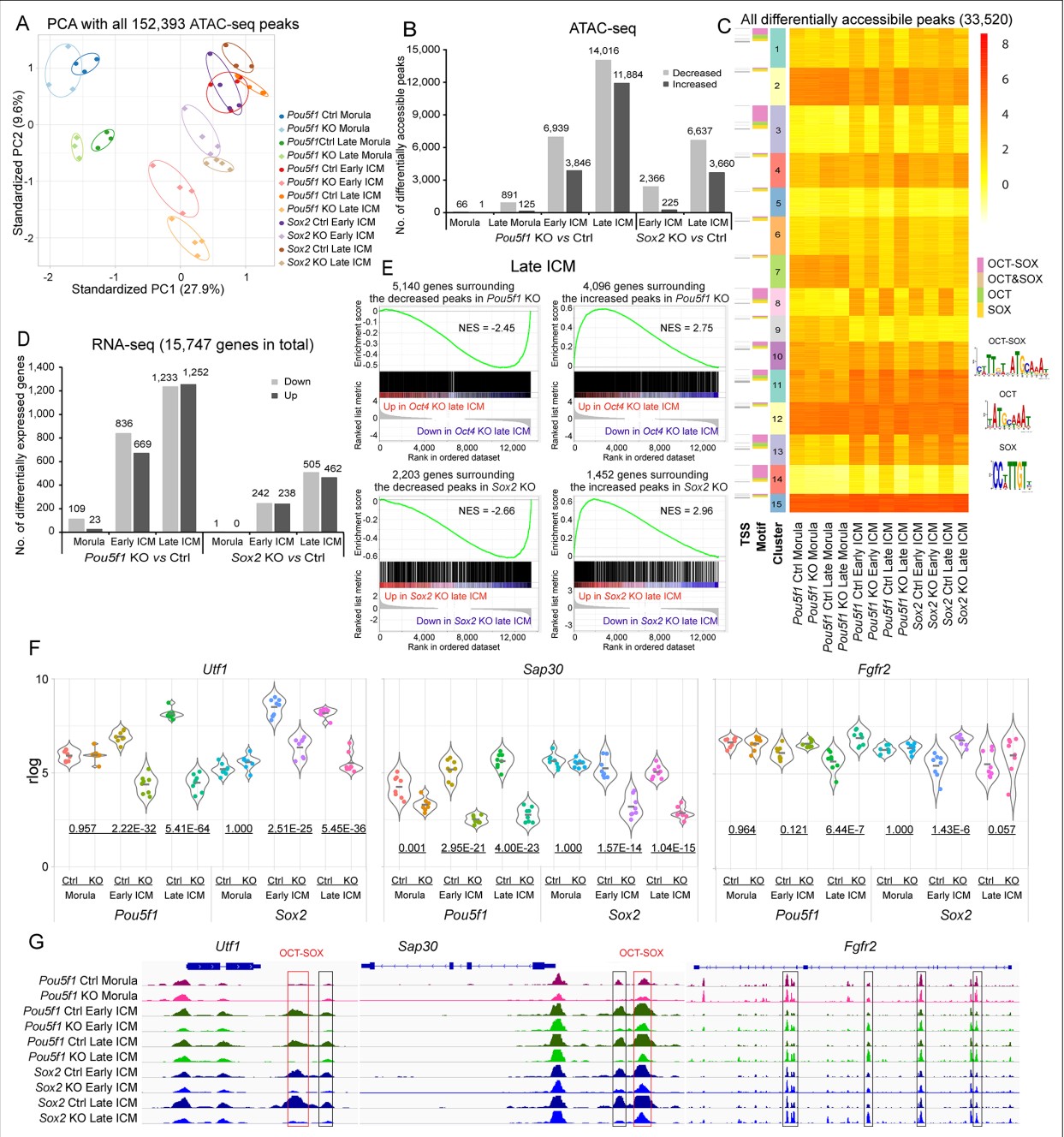

**Figure 2.** Oct4 and Sox2 regulate chromatin landscape and transcriptome in the inner cell mass (ICM). (**A**) Principal component analysis (PCA) plot of all the identified ATAC-seq peaks. (**B**) Number of differentially accessible ATAC-seq peaks in knockout (KO) vs Ctrl samples. Cutoff, adjusted p-value<0.05. (**C**) k-Means clustering of all the significantly differential ATAC-seq peaks in KO vs Ctrl in **B**. The heatmap is sorted by clusters, motifs, and transcription start sites (TSSs). Peaks located within 100 bp from TSS were considered as TSS peaks. OCT-SOX, OCT, or SOX motifs indicate that the peak contains the canonical OCT-SOX, OCT, or SOX motif, respectively, while OCT&SOX motif indicates that separate OCT and SOX motifs were discovered in one peak. Cutoff, adjusted p-value<0.05. (**D**) Number of differentially expressed genes in KO vs Ctrl samples. Cutoff, adjusted p-value<0.05 and log$_2$ fold change≥1. (**E**) Gene set enrichment analysis (GSEA) shows the correlation between significantly changed ATAC-seq peaks and the transcription of genes whose TSSs are located within 10 kb of the peak centers in *Pou5f1*- or *Sox2*-KO late ICMs. NES, normalized enrichment score. (**F**) Examples of down- and upregulated genes. The underlined numbers represent the adjusted p-values. (**G**) The ATAC-seq profiles surrounding the genes in E. Boxes mark the differentially accessible peaks, and red boxes specifically mark those with the OCT-SOX motif.

The online version of this article includes the following figure supplement(s) for figure 2:

**Figure supplement 1.** Quality check of the low-input ATAC-seq.

*Figure 2 continued on next page*

*Figure 2 continued*

**Figure supplement 2.** Quality check of the low-input RNA-seq.

**Figure supplement 3.** The correlation between chromatin accessibility and the transcription activity of surrounding genes.

At the transcriptional level, many known pluripotency genes, such as *Klf2*, *Etv5*, *Prdm14*, and *Pecam1*, were downregulated in the *Pou5f1*- and *Sox2*-KO ICM (*Figure 3C*). In contrast, *Gata3*, *Cdx2*, and *Eomes*, which are important for TE development and differentiation (*Ralston et al., 2010*; *Russ et al., 2000*; *Strumpf et al., 2005*), were upregulated in the *Pou5f1*- and Sox2-KO ICM. Interestingly, we observed that *Nanog*, *Esrrb*, and *Klf4* were significantly downregulated in the *Pou5f1*-KO ICM, whereas they were not or only slightly downregulated in the *Sox2*-KO ICM (*Figure 3C and D*). Downregulation of *Pecam1* was confirmed at the protein level (*Figure 3E*, *Figure 3—source data 1*, *Figure 3—source data 2*). Chromatin accessibility at its putative enhancers also decreased accordingly (*Figure 3F*). Oct4 and Sox2 activated the components of several epigenetic modifiers, such as *Ezh2* (PRC2 H3K27me2/3 methyltransferase) and *Sap30* (a component of mSin3A histone deacetylase complex) (*Figure 2F and G*; *Supplementary file 5*), suggesting their potential contribution to establishing the ICM-specific epigenetic status through regulation of the epigenetic modifiers.

Taken together, the above data show that Oct4 and Sox2 regulate a large number of TFs, epigenetic factors, and signaling pathways in the ICM.

## Oct4 and Sox2 co-activate their targets through putative OCT-SOX enhancers

Although there is a general consensus on the cooperative binding of Oct4 and Sox2 to the OCT-SOX composite motif, the principle of the cooperation still remains controversial in different scenarios (*Biddle et al., 2019*; *Chen et al., 2014*; *Friman et al., 2019*; *Li et al., 2019*; *Michael et al., 2020*). Therefore, we investigated how Oct4 and Sox2 regulate their target open chromatin regions in the ICM. As expected, the OCT-SOX, OCT, and SOX motifs were also among the most enriched motifs in the group of decreased peaks in *Pou5f1*- or *Sox2*-KO ICMs (*Figure 4A*). The motifs of other known pluripotency-related TFs, such as Klf2/4 and Esrrb, were also enriched at the decreased peaks. The elevated peaks were enriched for the GATA, TEAD, EOMES, and KLF motifs, but not for the OCT-SOX, OCT, or SOX motifs. Additionally, the gain of chromatin accessibility occurred at later stages compared to the loss of chromatin accessibility (*Figure 2B*). Thus, the increased peaks may not represent the direct targets of Oct4 or Sox2, but rather may be a secondary effect of the upregulated TE TFs, such as *Gata3*, *Tead4*, *Eomes*, and *Klf5/6*, or the disruption of inhibitory interactions between Oct4/Sox2 and TE TFs (*Niwa et al., 2005*; *Figure 3C*; *Supplementary file 5*). As a control, the CTCF motif was not enriched in either the decreased or increased peaks. We confirmed the enrichment of those motifs at distal peaks and concluded that the motif analysis was not biased by the small number of the TSS peaks in the dataset (*Figure 2—figure supplement 1A*, *Figure 4—figure supplement 1*).

To investigate the effect of Oct4 and Sox2 on the chromatin accessibility, we focused on the peaks in the ICM. It is worth noting that the top decreased peaks in the *Pou5f1*- and *Sox2*-KO early ICM were most enriched with the OCT-SOX, but not the OCT or SOX motifs (*Figure 4B*), which suggests that Oct4 and Sox2 could maintain open chromatin to a greater extent at peaks containing the OCT-SOX motif (hereafter referred to as OCT-SOX peaks). The KLF motif was more enriched at the moderately decreased peaks, indicating a possible secondary effect of downregulated *Klf2/4* (*Figure 3C and D*); alternatively, the depletion of Oct4 and Sox2 might lead to a decrease in DNA binding or the activity of Klf2/4 and other KLF TFs. In general, the accessibility of 8993 OCT-SOX peaks decreased in both the *Pou5f1*- and *Sox2*-KO early and late ICM (*Figure 4C*; *Figure 4—figure supplement 2*). Furthermore, majority of the decreased OCT-SOX peaks were shared in *Pou5f1*- and *Sox2*-KO late ICM (*Figure 4D*), suggesting that both TFs participate in keeping these peaks open. The known OCT-SOX enhancers of *Dppa3* and *Klf4*, which were enriched with the binding of Oct4 and Sox2, are shown as examples (*Figure 4E*).

The above results suggest that the putative enhancers containing the OCT-SOX motif are primary targets of Oct4 and Sox2 in the ICM.

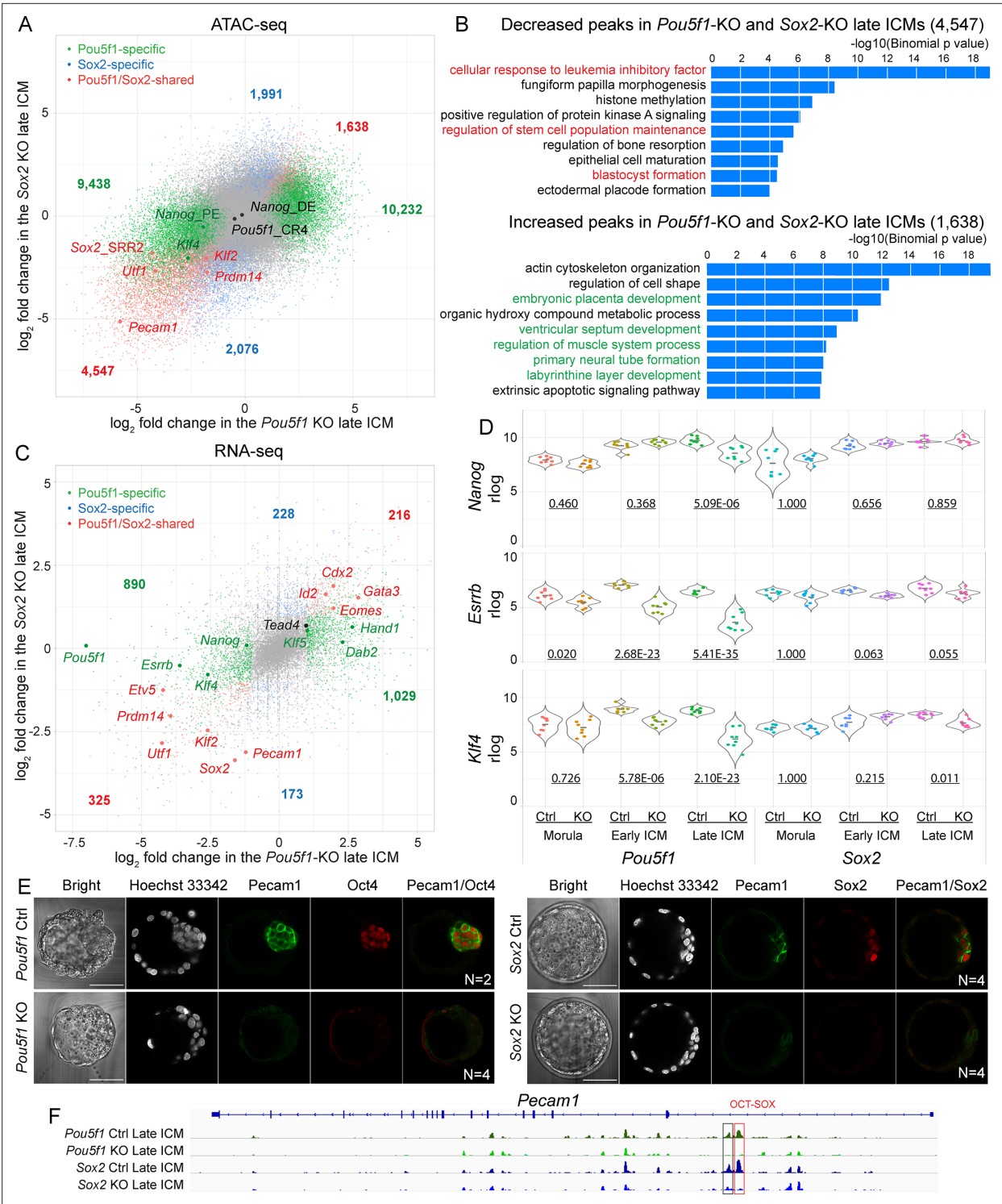

**Figure 3.** Oct4 and Sox2 activate epiblast (EPI)-specific genes and suppress trophectoderm (TE)-specific genes in inner cell mass (ICM). (**A**) A scatter plot shows the log₂ fold change of ATAC-seq signals in the *Pou5f1*-KO and *Sox2*-KO late ICM. Colored dots represent the significantly changed peaks. Cutoff, adjusted p-value<0.05. (**B**) GREAT ontology enrichment analysis of the significantly changed peaks shared in the *Pou5f1*-KO and *Sox2*-KO late ICM. Red terms are related to the pluripotency and preimplantation embryonic development, and green ones are related to the development of embryonic and extraembryonic lineages in the post-implantation embryos. (**C**) A scatter plot shows the log₂ fold changes of RNA-seq signals in the *Pou5f1*-KO and *Sox2*-KO late ICM. Cutoff, adjusted p-value<0.05 and log₂ fold change≥1. (**D**) Violin plots of RNA-seq rlog values for *Nanog, Esrrb,* and *Klf4* in the embryos. The underlined numbers represent the adjusted p-values. (**E**) Immunostaining of Pecam1 in the late blastocysts (E2.5+2 days).

*Figure 3 continued on next page*

*Figure 3 continued*

Scalebar, 50 µm. (**F**) The accessibility of putative enhancers around *Pecam1* in ICMs. Boxes mark the decreased peaks and red box marks the peak with the OCT-SOX motif.

The online version of this article includes the following source data and figure supplement(s) for figure 3:

**Source data 1.** Imaging data of embryos in panel E.

**Source data 2.** Imaging data of embryos in panel E.

**Figure supplement 1.** GREAT biological process enrichment analysis of ATAC-seq peaks specifically decreased in the *Pou5f1*-KO or *Sox2*-KO late inner cell masses (ICMs).

## Oct4 and Sox2 promote the developmental trajectory toward ICM

In early embryos, individual blastomeres are initially totipotent and indistinguishable before segregating into the ICM and TE. We subsequently investigated factors that influence the cellular trajectory toward the pluripotent ICM. Around 50% of the peaks and genes exhibiting decreased trend in *Pou5f1*- or *Sox2*-KO early ICMs were found to be activated in the Ctrl ICM during the natural development (*Figure 5A*). In alignment with earlier studies (*Guo et al., 2010*; *Wu et al., 2016*), a significant reorganization of open chromatin regions was observed upon the formation of the early ICM, resulting in the activation of 21,731 peaks (14.3%) (*Figure 5B*). The most prominently elevated peaks in the early ICM locate close to genes involved in cellular response to LIF, maintenance of stem cell population, and blastocyst formation, in accordance with the developmental stage (*Figure 5—figure supplement 1A*). These peaks were enriched for OCT, SOX, and OCT-SOX motifs (*Figure 5—figure supplement 1B*). However, when Oct4 and Sox2 were absent, the accessibility of the elevated peaks was decreased in the early ICM (*Figure 5C*). In alignment with the chromatin dynamics, the transcriptome also underwent substantial rearrangements in early ICMs, resulting in the upregulation of 1115 genes (7.1%) (*Figure 5D*). These 1115 genes were enriched with genes downregulated in the *Pou5f1*-KO and *Sox2*-KO early ICM (*Figure 5E*). These data indicate a critical role of Oct4 and Sox2 in initiating ICM-specific chromatin and transcriptional programs.

The network of naïve pluripotency is governed by a core regulatory circuit of TFs, including Oct4, Sox2, Nanog, Esrrb, and Klf4 (*Adachi et al., 2018*; *Ng and Surani, 2011*; *Young, 2011*). In particular, the activation of endogenous Sox2 represents a late-stage and deterministic event that triggers the fate toward pluripotency during reprogramming (*Buganim et al., 2012*). In mouse embryos, all the aforementioned TFs except Sox2 are highly expressed in the morula (*Figure 5—figure supplement 1C*); however, their expression does not seem sufficient to induce the mature pluripotent state at this stage. Therefore, we hypothesized that Sox2, in collaboration with Oct4, may activate the pluripotency-related genes in the early ICM. To investigate this hypothesis, we subjected the 1115 genes and their log$_2$fold change in *Sox2*-KO and *Pou5f1*-KO early ICMs to GSEA Wikipathway analyses (*Liao et al., 2019*). This analysis revealed an enriched downregulation of genes associated with PluriNetWork and ESC pluripotency pathways in *Sox2*-KO early ICMs (*Figure 5F*). In *Pou5f1*-KO early ICMs, although such enrichment was not observed, a significant downregulation of pluripotency-related genes was evident (*Figure 5—figure supplement 1D E*). For example, in the early ICM, Oct4 and Sox2 activate *Utf1* and *Il6st* (also known as *gp130*, receptor of the LIF/STAT pathway) through the open chromatin regions containing OCT-SOX motif (*Figures 2F, G and 5H*). Notably, the putative OCT-SOX enhancer of *Il6st* is enriched with Sox2 CUT&RUN signals. Additionally, Oct4 activated enzymes regulating the pyruvate metabolism (*Ldha, Me2, Pck2*, etc.) and glutathione metabolism (*Gstm1/2, Mgst2/3, Idh1*), consistent with previous studies highlighting Oct4's role in metabolism regulation (*Frum et al., 2013*; *Stirparo et al., 2021*). These findings indicate that Sox2 expression might function as a temporal regulator in the activation of putative OCT-SOX enhancers and a subset of pluripotency-related genes in the early ICM.

In summary, above data suggest that Oct4 and Sox2 play pivotal roles in promoting the ICM fate in early mouse embryos.

## Discussion

Oct4 and Sox2 are key TFs for preimplantation embryonic development across species (*Daigneault et al., 2018*; *Fogarty et al., 2017*; *Gao et al., 2022*; *Lee et al., 2013*; *Leichsenring et al., 2013*).

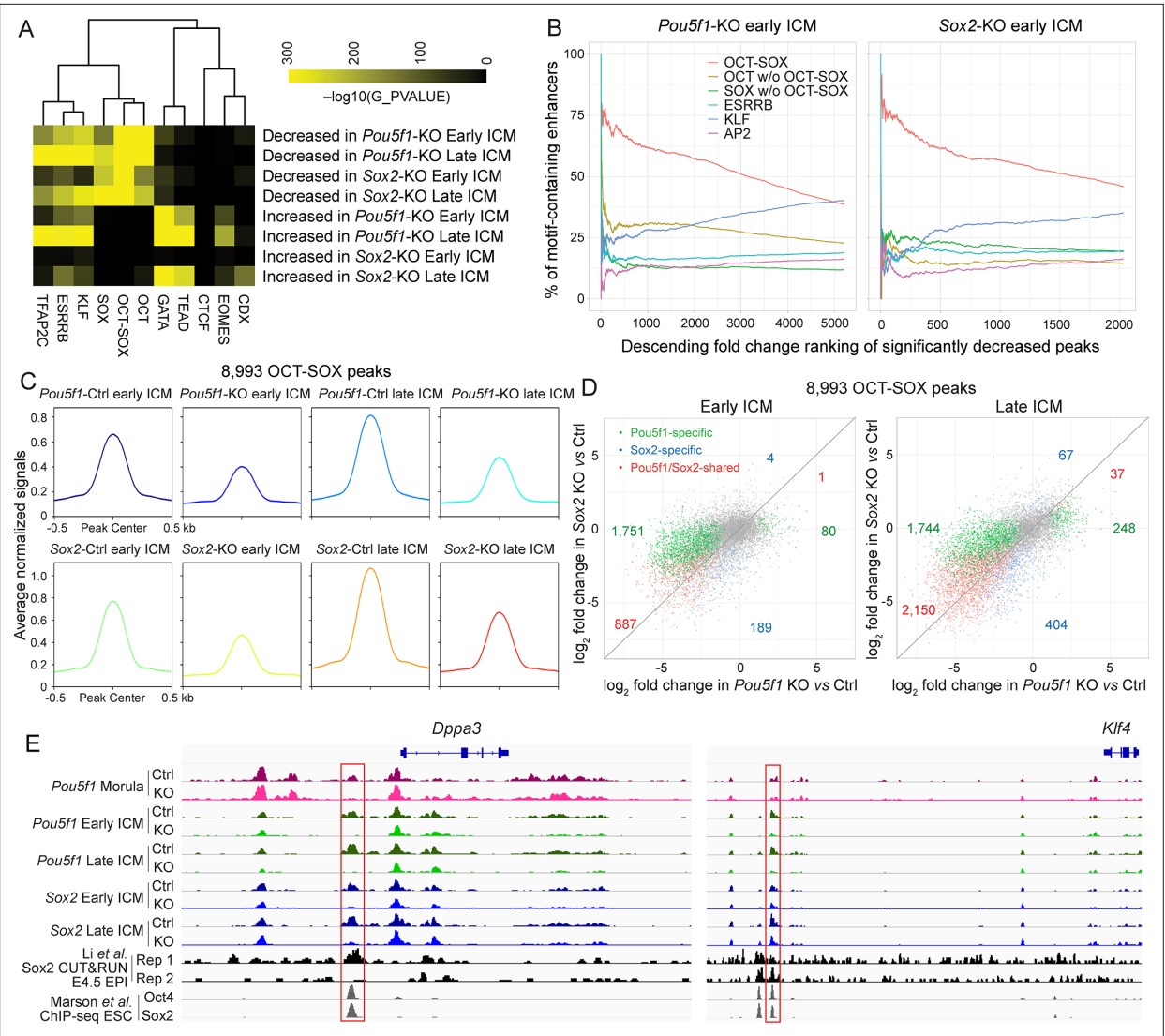

**Figure 4.** Oct4 and Sox2 activate OCT-SOX enhancers cooperatively and independently in inner cell mass (ICM). (**A**) Motif enrichment analysis of significantly changed ATAC-seq peaks in the *Pou5f1*-KO and *Sox2*-KO ICMs. (**B**) Occurrence of the motifs in the ranked peaks. The decreased enhancers were ranked by the fold reduction. The cumulative percentages of peaks containing at least one sequence for a given motif are plotted against ranks of peaks. (**C**) Average profiles of the 8993 OCT-SOX peaks across all the Ctrl and KO ICM samples. (**D**) Scatter plots show the log$_2$ fold change of 8993 OCT-SOX peaks in the *Pou5f1*- and *Sox2*-KO late ICMs. (**E**) The profiles of ATAC-seq in early embryos, Sox2 CUT&RUN in E4.5 EPI (***Li et al., 2023***), and Oct4 and Sox2 ChIP-seq in embryonic stem cells (ESCs) (***Marson et al., 2008***) around the known OCT-SOX enhancers of *Klf4* and *Dppa3*. The red boxes mark the OCT-SOX enhancers.

The online version of this article includes the following figure supplement(s) for figure 4:

**Figure supplement 1.** Motif enrichment analysis of significantly changed ATAC-seq peaks that are distal to transcription start sites (TSSs).

**Figure supplement 2.** MA plots show the log$_2$ fold change of the 8993 OCT-SOX peaks in *Pou5f1*- and *Sox2*-KO inner cell masses (ICMs).

However, because of the paucity of embryonic cells, the timing and mechanisms of their function in regulating the EPI lineage remain unclear. In this study, we explored the roles of Oct4 and Sox2 in mouse preimplantation embryos using transgenic models.

Previous reports suggested that Oct4 and Sox2 could direct 4-cell blastomeres toward ICM fate (***Plachta et al., 2011***; ***White et al., 2016***), and Oct4 facilitates chromatin opening at the 8-cell stage (***Lu et al., 2016***). Unexpectedly, our data showed that either *Pou5f1*-KO or *Sox2*-KO have minimal impact on global chromatin accessibility and transcription until the early blastocyst stage (***Figure 2A–D***). Oct4 is expressed at similar levels between morulae and ICMs; in contrast, Sox2 mRNA and protein levels are negligible in morulae, but are significantly upregulated in ICMs (***Figure 5—figure supplement***

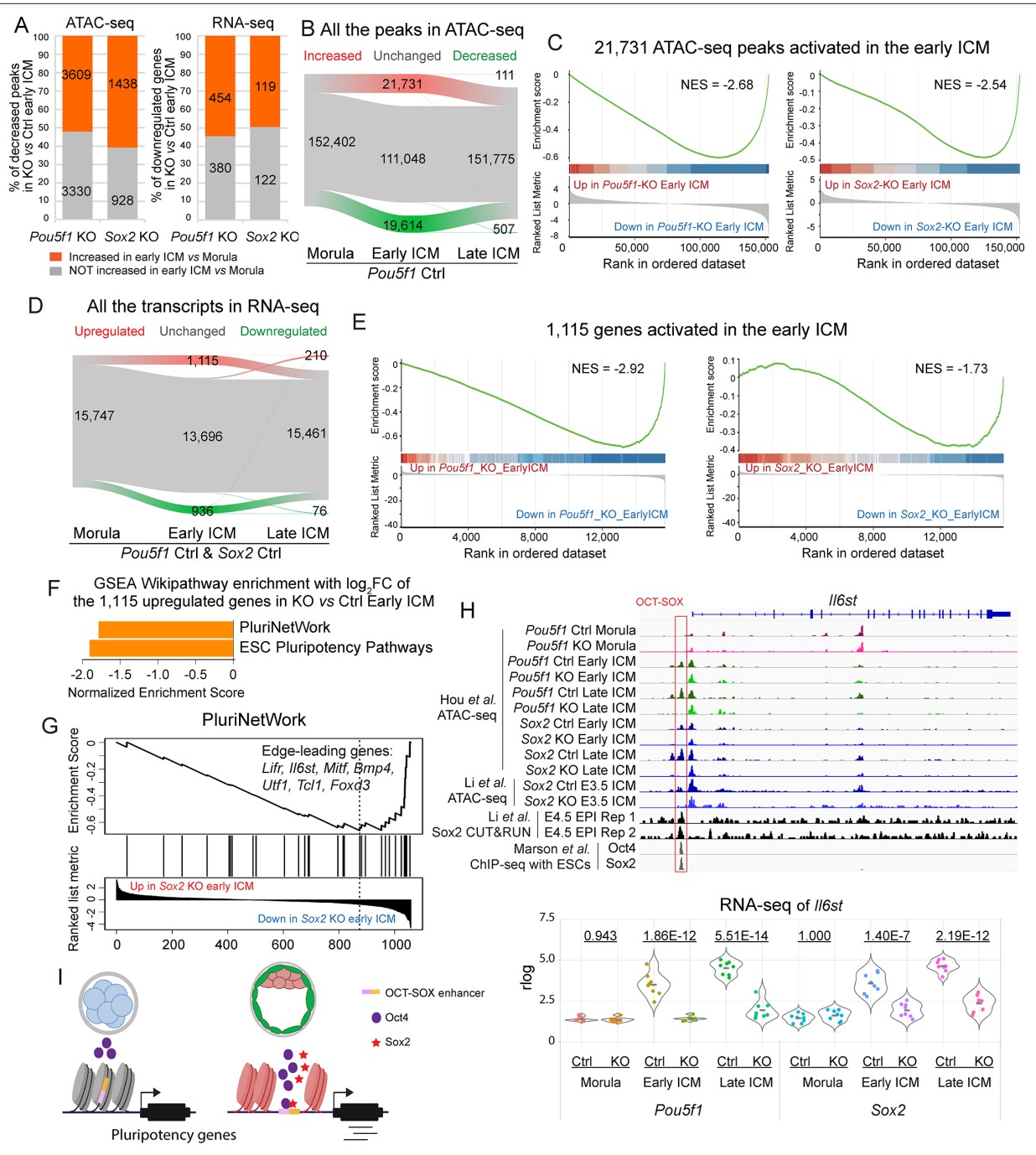

**Figure 5.** Oct4 and Sox2 promote the developmental trajectory from the morula to the inner cell mass (ICM). (**A**) Bar graphs show the dynamics of the decreased peaks (left) and genes (right) in *Pou5f1*- and *Sox2*-KO early ICMs from the morula to the early ICM. (**B and D**) Alluvial plots show the dynamics of chromatin accessibility (**B**) and transcriptome (**D**) from morula to late ICM in the Ctrl embryos. Green, gray, and red lines represent the decreased, unchanged, and increased peaks/genes, respectively. In D, only genes significantly up- or downregulated in both *Pou5f1* Ctrl and *Sox2* Ctrl embryos were considered as up- or downregulated genes, while all the rest were considered as unchanged genes. (C and E) Gene set enrichment analysis (GSEA) plots show the enrichment of the 21,731 ATAC-seq peaks (**C**) and 1115 genes (**E**) in *Pou5f1*-KO and *Sox2*-KO early ICMs. NES, normalized enrichment score. (**F**) The bar chart illustrates the GSEA Wikipathway enrichment in WebGestalt. The log$_2$ fold change values of the 1115 upregulated genes (D) in *Sox2*-KO vs Ctrl early ICM were used in this analysis. False discovery rate (FDR)≤0.05. FC, fold change. (**G**) GSEA enrichment plot of the term PluriNetWork in F. NES, normalized enrichment score. (**H**) Upper panel: IGV tracks displaying ATAC-seq and Sox2 CUT&RUN profiles (***Li et al., 2023***), along with ChIP-seq profiles of Oct4 and Sox2 in embryonic stem cells (ESCs) (***Marson et al., 2008***), centered around the genomic locus of *Il6st*. Red box marks the OCT-SOX enhancer. Lower panel: violin plot showing the rlog values of *Il6st*. The underlined numbers represent the adjusted p-values. (**I**) Model of the activation of pluripotency-related genes in the early embryos.

*Figure 5 continued on next page*

*Figure 5 continued*

The online version of this article includes the following figure supplement(s) for figure 5:

**Figure supplement 1.** The dynamics of chromatin accessibility and transcriptome from morula to inner cell mass (ICM).

---

*1C*; *Palmieri et al., 1994*; *Wicklow et al., 2014*). Therefore, in morulae, the function of Oct4 is likely limited by the insufficient levels of Sox2 and possibly other necessary factors. Another plausible limiting aspect could be epigenetic constraints. The epigenetic state and three-dimensional structure of the chromatin in the morula differ from those in the ICM, which may hinder Oct4 function (*Du et al., 2017*; *Wang et al., 2018*; *Wang et al., 2014*; *Zhang et al., 2016*).

Endogenous Oct4 is activated prior to Sox2 in both embryogenesis and reprogramming. In mouse embryos, Oct4 is initially highly expressed in all blastomeres from the 8-cell stage and later restricted to the EPI, whereas Sox2 is repressed by the Hippo pathway in early stages and is specifically expressed in the EPI (*Frum et al., 2019*; *Guo et al., 2010*; *Wicklow et al., 2014*). In reprogramming, the activation of endogenous Sox2 occurs later than the endogenous Oct4 and signifies the final phase of iPSC generation (*Buganim et al., 2012*). In both scenarios, the expression of Sox2 coincides with the setup of the pluripotent state. Our data show that *Sox2*-KO ICMs fail to fully activate the pluripotent program (*Figure 3C*). Recently, we reported that the loss of Sox2 might contribute to decreased developmental potential of pluripotent cells upon priming (*MacCarthy et al., 2024*). Above studies suggest that Sox2 plays a critical role in regulating the spatial-temporal full onset of pluripotency-related genes.

Loss of Oct4 and Sox2 impairs embryonic pluripotency, as evidenced by the inability of KO embryos to give rise to ESCs or contribute to the embryonic part in chimera assays (*Avilion et al., 2003*; *Nichols et al., 1998*; *Wu et al., 2013*). To date, only a few of pluripotency-related genes have been found downstream of Oct4 and Sox2 in mouse embryos. Here, we found that a large number of pluripotency-related genes were downregulated in *Pou5f1*-KO ICM, contrasting with previous reports which suggested that most pluripotency-related genes were maintained (*Frum et al., 2013*; *Le Bin et al., 2014*; *Stirparo et al., 2021*; *Wu et al., 2013*). This discrepancy may be due to differences in the experimental design. First, unlike previous study that used the whole blastocysts (*Frum et al., 2013*), we isolated an EPI cell population through immunosurgery and short-term inhibition of MEK while it did not alter Oct4/Sox2 target open chromatin regions and genes (*Nichols et al., 2009*; *Figure 1—figure supplement 2*; *Figure 2—figure supplement 1B–D*, *Figure 1—figure supplement 2B–D*). Second, we removed *Pou5f1* exons 2–5 (the whole DNA-binding POU domain), resulting in a complete ablation of Oct4 function (*Figure 1A*). In contrast, majority of the previous studies removed only exon 1 (*Frum et al., 2013*; *Wu et al., 2013*), which encodes for N-terminal transactivation domain. The remaining exons 2–5 might be translated and could retain partial function. Finally, unlike the previous study that used conventional KO (maternal-WT zygotic-KO) embryos (*Stirparo et al., 2021*), we used maternal-zygotic-KO embryos (*Figure 1C*). The residual levels of maternal Oct4 could have potentially rescued the KO.

Preimplantation embryos undergo dynamic epigenomic reprogramming, governed by multiple epigenetic modifiers. For example, *Ezh2* is required for the establishment and maintaining global H3K27me3, while *Uhrf1* maternal-KO embryos fail to form healthy morula due to disrupted DNA methylation and histone modifications (*Cao et al., 2019*; *Maenohara et al., 2017*; *Puschendorf et al., 2008*). Our data show that Oct4 and Sox2 activate components of epigenetic modifiers like *Ezh2*, *Uhrf1*, and *Sap30* (*Figure 2F and G*; *Figure 2—figure supplement 1E*; *Supplementary file 5*), suggesting their role in regulating epigenetic status.

While the cooperation of Oct4 and Sox2 on the OCT-SOX enhancers is widely acknowledged, there are still debates regarding their precise roles (*Biddle et al., 2019*; *Chen et al., 2008*; *Han et al., 2022*; *Li et al., 2019*; *Michael et al., 2020*; *Velychko et al., 2019*). In this study, most OCT-SOX peaks are more affected in *Pou5f1*-KO than in *Sox2*-KO ICMs, indicating that Oct4 plays a more important role than Sox2 in activating putative OCT-SOX enhancers in vivo. This observation aligns with a previous study showing that the forced expression of Oct4 can rescue pluripotency in *Sox2*-null ESCs (*Masui et al., 2007*). We believe that this outcome is unlikely to be attributed to compensation from other members in the Sox family. Sox1, Sox3, Sox15, and Sox18 are promising candidates as they can either rescue the *Sox2*-KO mESCs or replace Sox2 in reprogramming (*Nakagawa et al., 2008*; *Niwa et al., 2016*). However, our analysis reveals that *Sox1*, *Sox3*, and *Sox18* are expressed at extremely low levels

in wildtype and *Sox2*-KO embryos (**Supplementary file 5**), suggesting that they are unlikely to be able to fulfill Sox2's role (**Deng et al., 2014**). Sox15 exhibits indistinguishable expression levels in all three cell types at the blastocyst stage. Furthermore, *Sox15* KO mice displays normal health and fertility (**Lee et al., 2004**). Nonetheless, we cannot completely rule out the possibility of compensation by Sox15 in *Sox2*-KO embryos.

Our results highlight the crucial role of Oct4 and Sox2 in establishing the transcriptome and chromatin state in the pluripotent EPI. At the blastocyst stage, Oct4 and Sox2 work together to open the putative OCT-SOX enhancers and activate the pluripotency-related genes (**Figure 5I**). The absence of Sox2 and other factors likely limits the function of Oct4 in morulae. However, the upstream factors driving the activation of the core pluripotency regulatory circuitry remain unknown. Further studies are needed to deepen our understanding of the molecular mechanisms governing the embryonic pluripotency program and the early lineage segregation.

## Materials and methods

### Mice

To generate ESCs carrying a floxed *Pou5f1* allele (*Pou5f1^flox^*), the construct containing floxed *Pou5f1* exons 2–5 and a promoter-less FRT-IRES-βgeo-pA cassette was electroporated into germline-competent Acr-EGFP ESCs. Clones were screened for homologous recombination and transiently transfected with an FLP expression vector to remove the FRT cassette. To generate ESCs carrying a mutant *Pou5f1* allele linked to *mKO2* (*Pou5f1^mKO2^*), Acr-EGFP ESCs were targeted with the construct containing a PGK-pacΔtk-P2A-mKO2-pA cassette 3.6 kb upstream of the *Pou5f1* TSS and a promoter-less FRT-SA-IRES-hph-P2A-Venus-pA cassette in *Pou5f1* intron 1.

To generate ESCs carrying the floxed *Sox2* allele (*Sox2^flox^*), the construct containing the 5' loxP site 1.9 kb upstream of *Sox2* TSS and a promoter-less FRT-IRES-Neo-pA cassette followed by the 3' loxP site in the 3' UTR of *Sox2* was electroporated into Acr-EGFP ESCs. Clones were screened for homologous recombination and transiently transfected with an FLP expression vector to remove the FRT cassette. To generate ESCs carrying a conditional allele of *Sox2* linked to *EGFP* (*Sox2^EGFP^*), Acr-EGFP ESCs were targeted with the construct containing the 5' loxP site 1.9 kb upstream of *Sox2* TSS, and a promoter-less FRT-IRES-Neo-pA cassette, the 3' loxP site, and a PGK-EGFP-pA cassette in the 3' UTR of *Sox2*.

Chimeric mice were generated by morula aggregation and heterozygous mice were obtained through germline transmission. The surrogate morulae were obtained by breeding B6C3 F1 females with CD1 males. The *Zp3-Cre* transgenic mice were used to generate maternal KO alleles.

All mouse experiments and husbandry were conducted at the animal facility of the Max Planck Institute for Molecular Biomedicine, in compliance with the institute's animal welfare guidelines. The study was performed under an approved protocol for animal care and use (84-02.04.2014.A239).

### Embryo collection

All the embryos for RNA-seq and ATAC-seq were collected from female mice superovulated with PMSG and hCG. Early and late morula samples were collected at E2.75 and E3.25, respectively. Early and late ICM samples were collected at E2.5 and cultured in KSOM with MEKi (PD0325901, 1 µM) until early and late blastocyst stages, respectively.

### ICM isolation

Zona pellucida was removed by brief incubation in prewarmed acidic Tyrode's solution. ICM was obtained by immunosurgery. Briefly, blastocysts were incubated in 20% rabbit anti-mouse whole serum (Sigma-Aldrich) in KSOM at 37°C, 5% $CO_2$ for 15 min, followed by three rinses in M2. Afterward, embryos were incubated in 20% guinea pig complement serum (Sigma-Aldrich) in KSOM at 37°C, 5% $CO_2$ for 15 min, followed by three rinses in M2. In the end, ICM was isolated by repetitive blowing in mouth pipette to remove the debris of dead trophectoderm cells.

### Immunofluorescence

Immunofluorescence was performed using the following primary antibodies with dilutions: mouse monoclonal anti-Oct4 (sc-5279, Santa Cruz), 1:1000; goat anti-Sox2 (GT15098, Neuromics), 1:300;

rabbit anti-mKO2 (PM051, MBL), 1:1000; rabbit anti-GFP (ab290, Abcam), 1:500; goat anti-Pecam1 (AF3628, R&D Systems), 1:300; rat monoclonal anti-Nanog (14-5761-80, Thermo Fisher Scientific), 1:100; goat anti-Sox17 (AF1924, R&D Systems), 1:1000 of 0.2 mg/ml.

## ATAC-seq and data analysis

ATAC-seq libraries were prepared as previously described with some modifications. 20–40 morulae or ICMs were pooled and lysed in the lysis buffer (10 mM Tris·HCl, pH = 7.4; 10 mM NaCl; 3 mM MgCl$_2$; 0.15% Igepal CA-630; 0.1% Tween-20) for 10 min on ice. We added 0.1% Tween-20 to the lysis buffer as it greatly reduced reads mapped to mitochondrial DNA. Lysed embryos were briefly washed, collected in 2.0 µl PBS containing 0.1 mg/ml polyvinyl alcohol, and transferred to 3.0 µl Tn5 transposome mixture (0.25 µl Tn5 transposome, 2.5 µl tagmentation buffer, Illumina, FC-121-1030; 0.25 µl H$_2$O). The samples were incubated in 37°C water bath for 30 min. Tagmentation was stopped by adding 2.0 µl 175 mM EDTA and incubated at 50°C for 30 min. Excess EDTA was quenched by 2.0 µl 160 mM MgCl$_2$. The libraries were amplified in the following reaction: 9.0 µl Transposed DNA, 10.0 µl NEBNext High-Fidelity 2×PCR Master Mix (New England Biolabs, M0541S), 0.25 µl 100 µM PCR Index 1, 0.25 µl 100 µM PCR Index 2, and 0.5 µl H$_2$O. The sequences of indexes 1 and 2 are in *Supplementary file 2*. The PCR program is as the following: 72°C for 5 min; 98°C for 30 s; 16 thermo-cycles at 98°C for 10 s, 63°C for 30 s, and 72°C for 1 min, followed by 72°C 5 min. Amplified libraries were purified twice with 1.2× AMPure XP beads. The sequencing was performed on the NextSeq 500 system with paired-end 75 bp.

Sequence reads were trimmed for adapter sequences using SeqPurge and trimmed reads no shorter than 20 bases were aligned to the mm10 mouse reference genome using Bowtie2 with a maximum fragment size of 2000. Duplicated reads were removed using Picard MarkDuplicates (*Broad Institute, 2019*) and only reads uniquely mapped to the standard chromosomes except chrY and chrM (mitochondrial genome) with mapping quality of at least 30 were used for the following analysis. Reads on the positive and negative strands were shifted +4 bp and −5 bp, respectively. Peak calling was performed using MACS2 with the following parameters: `--keep-dup` all -g mm `--nomodel --shift` −50 `--extsize` 100 -B `--SPMR`. Peaks that overlap with the blacklisted regions (https://sites.google.com/site/anshulkundaje/projects/blacklists; https://sites.google.com/site/atacseqpublic/atac-seq-analysis-methods/mitochondrialblacklists-1) and satellite repeats (Repeat-Masker) were removed. Significant peak summits (q-value≤0.001) from biological replicates were merged for each sample group using BEDTools with a maximum distance between two summits to be merged of 200 bp. Only merged peak summits that overlap with summits from at least two replicates were retained. Reads per million (RPM)-normalized pileup signals in the bedGraph format were converted into bigWig files using the UCSC bedGraphToBigWig tool. The bigWig files for the average signal of biological replicates were generated using the UCSC bigWigMerge tool. The heat-maps of RPM-normalized pileup signals around the TSSs of GENCODE (vM23) protein-coding genes were generated using deepTools.

For count-based analysis of transposon insertion events, merged peak regions were generated by combining the adjacent peak summits of all embryo groups and selecting 200 bp regions around the centers of each merged summit. 5′ ends of shifted read alignment on both the positive and negative strands were considered as transposon insertion sites. The number of transposon inser-tions in each merged peak region was counted for each sample using BEDTools and normalized and transformed to log$_2$ scale using the rlog function of the Bioconductor package DESeq2 and significantly differential peaks were identified with an adjusted p-value cutoff of 0.05. Peaks located within 100 bp from TSSs of GENCODE (vM23) genes were considered as TSS-proximal peaks. The presence of TF-binding motifs in merged peaks was investigated using FIMO with the default cutoff (p-value<10$^{-4}$).

The following JASPAR motifs were used: MA0036.3 GATA2, MA0139.1 CTCF, MA0141.3 ESRRB, MA0142.1 Pou5f1-Sox2, MA0143.3 Sox2, MA0524.2 TFAP2C, MA0599.1 KLF5, MA0792.1 POU5F1B, MA0800.1 EOMES, MA0808.1 TEAD3, and MA0878.1 CDX1. Significantly enriched motifs in each peak cluster were identified from using PscanChIP (*Zambelli et al., 2013*). Functional annotation of gene sets associated with ATAC-seq peaks was performed using GREAT (*McLean et al., 2010*) with default settings.

## RNA-seq and data analysis

Single-embryo RNA-seq was performed as previously described with some modifications (**Kurimoto et al., 2007**, **Nakamura et al., 2015**). 2493 copies of ERCC RNA Spike-In (Ambion) were added to each single-embryo sample. To remove PCR duplicates, we used R2SP-UMI-d(T)24 primer, instead of V1d(T)24 primer, for reverse transcription (RT). To reduce byproducts derived from the RT primer, the poly(A) tailing reaction was performed only for 1 min at 37°C. cDNA was amplified with V3d(T)24 and R2SP primers and Terra PCR Direct Polymerase (Clontech) for 16 cycles. The cDNA was further amplified for four cycles with NH2-V3d(T)24 and NH2-R2SP primers, and purified thrice with 0.6× AMPure XP beads. The 3' end enriched libraries were constructed using KAPA HyperPlus Library Preparation Kit (VWR International, KK8513) with 6 cycles of PCR. All the oligos used in the RT and library amplification are in **Supplementary file 3**. The sequencing was performed on the NextSeq 500 system with single-end 75 bp.

UMI sequence of each read was extracted from FASTQ files for Read 2 using UMI-tools. Poly-A sequences were trimmed from the 3' end of Read 1 using Cutadapt (https://cutadapt.readthedocs.org/). The trimmed Read 1 sequences were aligned to the mm10 mouse reference genome using TopHat2 with GENCODE (vM23) transcripts as a transcriptome reference and with default parameters. The reads were also mapped to ERCC reference sequences using Bowtie2 with default parameters. To confirm the genotype of *Pou5f1*- and *Sox2*-KO embryos, we also mapped the reads to the PGK-pacΔtk-P2A-mKO2-hGHpA or PGK-EGFP-hGHpA cassette. The reads mapped to the genome were assigned to GENCODE transcripts using featureCounts. The number of unique transcripts assigned to each GENCODE gene was calculated using TRUmiCount. The transcript counts were normalized and transformed to $\log_2$ scale using the rlog function of the Bioconductor DESeq2 package. Differential expression analysis was performed on genes that were detected in at least half of the samples in at least one stage using DESeq2 and differentially expressed genes were identified with an adjusted p-value<0.05 and $\log_2$ fold change≥1.

## GSEA

GSEA (**Subramanian et al., 2005**) was performed to determine whether genes close to the selected ATAC-seq peaks are enriched in genes differentially expressed in *Pou5f1*- or *Sox2*-KO embryos. The ATAC-seq peaks were annotated to Ensembl protein-coding genes whose TSSs are located within 10 kb of the peak centers using the R package ChIPpeakAnno (**Zhu et al., 2010**). False discovery rate (FDR) was estimated by gene set permutation tests. Heatmaps were generated from normalized enrichment scores (NESs) for gene sets with FDR≤0.1.

## Acknowledgements

We appreciate the invaluable assistance from Ingrid Gelker, Claudia Ortmeier, David Obridge, Martina Sinn, and the animal facility. Special thanks go to Dong Han, Rui Fan, Eva Kutejova, and Erik Tolen for their insightful discussions. We acknowledge the generous support from Anika Witten, Christoph Bartenhagen, and Carolin Walter of the Core Facility Genomics at the University of Muenster for their support with sequencing. Additionally, we would like to express our gratitude to Shixue Gou and Hui Zhang from the Guangzhou National Laboratory for their kind suggestions on RNA-seq data analysis and manuscript revisions, respectively. This work was funded by the Max Planck Society and the White Paper Project 'Animal testing in the Max-Planck-Society'.

## Additional information

### Funding

| Funder | Grant reference number | Author |
| --- | --- | --- |
| Max Planck Society | the White Paper Project "Animal testing in the Max-Planck-Society" | Hans R Scholer |

| Funder | Grant reference number | Author |
|---|---|---|

The funders had no role in study design, data collection and interpretation, or the decision to submit the work for publication. Open access funding provided by Max Planck Society

## Author contributions

Yanlin Hou, Conceptualization, Data curation, Formal analysis, Validation, Visualization, Methodology, Writing – original draft, Writing – review and editing; Zhengwen Nie, Sandra Heising, Methodology; Qi Jiang, Software, Visualization; Sergiy Velychko, Ivan Bedzhov, Writing – review and editing; Guangming Wu, Supervision, Methodology, Project administration; Kenjiro Adachi, Conceptualization, Data curation, Software, Formal analysis, Supervision, Validation, Investigation, Visualization, Methodology, Project administration, Writing – review and editing; Hans R Scholer, Resources, Supervision, Funding acquisition, Project administration, Writing – review and editing

## Author ORCIDs

Yanlin Hou  http://orcid.org/0009-0005-0582-8303
Sergiy Velychko  https://orcid.org/0000-0002-6227-3966
Guangming Wu  http://orcid.org/0000-0003-1923-7609
Kenjiro Adachi  https://orcid.org/0009-0003-0161-1508
Hans R Scholer  https://orcid.org/0000-0002-7422-8847

## Ethics

All mouse experiments and husbandry were performed at the mouse facility of Max Planck Institute for Molecular Biomedicine. Animal handling was in accordance with Max Planck Institute animal protection guidelines. A protocol for animal handling and maintenance for this study was approved by the Landesamt fur Natur, Umwelt und Verbraucherschutz Nordrhein-Westfalen. (84-02.04.2014.A239).

Reviewer #1 (Public review): https://doi.org/10.7554/eLife.100735.3.sa1
Reviewer #2 (Public review): https://doi.org/10.7554/eLife.100735.3.sa2
Author response https://doi.org/10.7554/eLife.100735.3.sa3

# Additional files

## Supplementary files

Supplementary file 1. Coordinates for the marked well-known enhancers in *Figure 3A*.

Supplementary file 2. Oligos for ATAC-seq library preparation.

Supplementary file 3. Oligos for RNA-seq library preparation.

Supplementary file 4. DESeq2 of ATAC-seq peaks.

Supplementary file 5. DESeq2 of RNA-seq.

MDAR checklist

## Data availability

ATAC-seq and RNA-seq data have been deposited at GEO under GSE264614 and GSE264615, respectively. Published high-throughput sequencing datasets used in this manuscript are listed as follows: ATAC-seq and CUT&RUN of early embryos, GSE203194; scRNA-seq of early embryos, GSE203194 and GSE159030; Oct4/Sox2/Nanog ChIP-seq of mESCs, GSE11724; H3K27ac ChIP-seq of mESCs, GSE27844.

The following datasets were generated:

| Author(s) | Year | Dataset title | Dataset URL | Database and Identifier |
|---|---|---|---|---|
| Hou Y, Adachi K, Wu G | 2024 | Emerging cooperativity between Oct4 and Sox2 governs the pluripotency network in mouse early embryos [ATAC-seq] | https://www.ncbi.nlm.nih.gov/geo/query/acc.cgi?acc=GSE264614 | NCBI Gene Expression Omnibus, GSE264614 |
| Hou Y, Adachi K, Wu G | 2024 | Emerging cooperativity between Oct4 and Sox2 governs the pluripotency network in mouse early embryos [RNA-seq] | https://www.ncbi.nlm.nih.gov/geo/query/acc.cgi?acc=GSE264615 | NCBI Gene Expression Omnibus, GSE264615 |

The following previously published datasets were used:

| Author(s) | Year | Dataset title | Dataset URL | Database and Identifier |
|---|---|---|---|---|
| Li L, Lai F, Xie W | 2023 | Multifaceted SOX2-chromatin interaction underpins pluripotency progression in early embryos | https://www.ncbi.nlm.nih.gov/geo/query/acc.cgi?acc=GSE203194 | NCBI Gene Expression Omnibus, GSE203194 |
| Stirparo GG | 2021 | Oct4 induces embryonic pluripotency via Stat3 signalling and metabolic mechanisms | https://www.ncbi.nlm.nih.gov/geo/query/acc.cgi?acc=GSE159030 | NCBI Gene Expression Omnibus, GSE159030 |
| Young RA | 2008 | Connecting microRNA genes to the core transcriptional regulatory circuitry of embryonic stem cells | https://www.ncbi.nlm.nih.gov/geo/query/acc.cgi?acc=GSE11724 | NCBI Gene Expression Omnibus, GSE11724 |
| Young RA | 2012 | Enhancer Decommissioning by LSD1 During Embryonic Stem Cell Differentiation | https://www.ncbi.nlm.nih.gov/geo/query/acc.cgi?acc=GSE27844 | NCBI Gene Expression Omnibus, GSE27844 |

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
