## [Editor Report · eLife Assessment]

This study presents a **valuable** finding on how the interplay between transcription factors SOX2 and OCT4 establishes the pluripotency network in early mouse embryos. The evidence supporting the claims of the authors is **solid**, although inclusion of additional omics data would further strengthen the study. The work will be of interest to biologists working on embryonic development and gene regulation.

---

## [Referee Report · Reviewer #1 (Public review)]

Summary:

Numerous mechanism and structural studies reported the cooperative role of Oct4 and Sox2 during the establishment of pluripotency during reprogramming. Due to the difficulty in sample collection and RNA-seq with low-number cells, the precise mechanisms remain in early embryos. This manuscript reported the role of OCT4 and SOX2 in mouse early embryos using knockout models with low-input ATAC-seq and RNA-seq. Compared to the control, chromatin accessibility and transcriptome were affected when Oct4 and Sox2 were deleted in early ICM. Specifically, decreased ATAC-seq peaks showed enrichment of Motifs of TF such as OCT, SOX, and OCT-SOX, indicating their importance during early development. Moreover, by deep analysis of ATAC-seq and RNA-seq data, they found Oct4 and Sox2 target enhancer to activate their downstream genes. In addition, they also uncovered the role of OS during development from the morula to ICM, which provided the scientific community with a more comprehensive understanding.

Strengths:

On the whole, the manuscript is innovative, and the conclusions of this paper are mostly well supported by data.

Weaknesses:

Major Points:

(1) In Figure 1, a more detailed description of the knockout strategy should be provided to clarify itself. The knockout strategy in Fig1 is somewhat obscure, such as how is OCT4 inactivated in Oct4mKO2 heterozygotes. As shown in Figure 1, the exon of OCT4 is not deleted, and its promoter is not destroyed. Therefore, how does OCT4 inactivate to form heterozygotes?

(2) Is ZP 3-Cre expressed in the zygotes? Is there any residual protein?

(3) What motifs are enriched in the rising ATAC-seq peaks after knocking out of OCT4 and SOX2?

(4) The ordinate of Fig4c is lost.

(5) Signals of H3K4me1, H3K27ac, and so on are usually used to define enhancers, and the loci of enhancers vary greatly in different cells. In the manuscript, the authors defined ATAC-seq peaks far from the TSS as enhancers. The definition in this manuscript is not strictly an enhancer.

(6) If Oct4 and Sox2 truly activate sap 30 and Uhrf 1, what effect does interfering with both genes have on gene expression and chromatin accessibility？

Comments on revisions:

The authors have addressed my concerns so I am fine with revision in principle.

---

## [Referee Report · Reviewer #2 (Public review)]

In this manuscript, Hou et al. investigate the interplay between OCT4 and SOX2 in driving the pluripotent state during early embryonic lineage development. Using knockout (KO) embryos, the authors specifically analyze the transcriptome and chromatin state within the ICM-to-EPI developmental trajectory. They emphasize the critical role of OCT4 and the supportive function of SOX2, along with other factors, in promoting embryonic fate. Although the paper presents high-quality data, several key claims are not well-supported, and direct evidence is generally lacking.

Comments on revisions:

The authors have addressed many of the concerns raised in the initial review and provided alternative analytical approaches to address the relevant questions in this revision. Some of these are useful; however, they have not fully addressed one critical point.

In my original critique, I noted that the maternal KO might not be suitable as a control, given that there is no significant phenotypic difference between the maternal-only KO and the maternal-zygotic KO. While we did not dispute the molecular differences presented in Figure 2, so how the authors conclude in the Response "embryos with a maternal KO or zygotic heterozygous KO of Oct4 or Sox2 show no noticeable ... molecular difference (Figure 2-figure supplement 4A)"? The authors should recheck whether this is a typographical error or a valid statement.

Additionally, I recommend the removal of phrases such as "absolutely priority" and "pivotal" throughout the manuscript, as these terms are overly assertive without sufficient supporting evidence.

---

## [Author Response]

The following is the authors’ response to the current reviews.

**Public Reviews:**

**Reviewer #1 Comments on revisions:**
The authors have addressed my concerns so I am fine with revision in principle.

Thank you for taking the time to review our work and for your thoughtful feedback. We’re glad to hear that your concerns have been addressed.

**Reviewer #2 Comments on revisions:**
The authors have addressed many of the concerns raised in the initial review and provided alternative analytical approaches to address the relevant questions in this revision. Some of these are useful; however, they have not fully addressed one critical point.In my original critique, I noted that the maternal KO might not be suitable as a control, given that there is no significant phenotypic difference between the maternal-only KO and the maternal-zygotic KO. While we did not dispute the molecular differences presented in Figure 2, so how the authors conclude in the Response "embryos with a maternal KO or zygotic heterozygous KO of Oct4 or Sox2 show no noticeable ... molecular difference (Figure 2-figure supplement 4A)"? The authors should recheck whether this is a typographical error or a valid statement.Additionally, I recommend the removal of phrases such as "absolutely priority" and "pivotal" throughout the manuscript, as these terms are overly assertive without sufficient supporting evidence.

We sincerely appreciate the reviewer’s feedback and would like to take this opportunity to provide further clarification, as there might have been a misunderstanding.

We respectfully disagree with the reviewer’s statement that “there is no significant phenotypic difference between the maternal-only KO and the maternal-zygotic KO.” Based on privious publications, there is clear evidence that maternal-zygotic KO embryos exhibit significant defects: they fail to form a healthy primitive endoderm, are unable to give rise to embryonic stem cells (ESCs) in vitro, and die shortly after implantation (Frum et al., Dev Cell 2013; Wu et al., Nat Cell Biol 2013; Le Bin et al., Development 2014; Wicklow et al., PLoS Genet 2014). In contrast, maternal-only KO embryos develop as healthy as wild-type (WT) embryos and do not display any of these phenotypic abnormalities. We believe that this distinction validates our use of maternal KO embryos as proper controls in our experiments.

To address the reviewer’s concerns and ensure clarity, we have also revised the following statement in the manuscript.

Original manuscript: “Mouse embryos with a maternal KO or zygotic heterozygous KO of either factor show no noticeable phenotype or molecular difference (Figure 2-figure supplement 4A) (Avilion et al., 2003; Frum et al., 2013; Kehler et al, 2004; Nichols et al., 1998; Wicklow et al., 2014; Wu et al., 2013).”

Revised manuscript: “Maternal KO embryos (circles in Figure 2—figure supplement 4A) clustered together with wildtype embryos (triangles and squares) in the PCA analysis, consistent with previous studies reporting no observable phenotype in maternal KO embryos (Avilion et al., 2003; Frum et al., 2013; Kehler et al, 2004; Nichols et al., 1998; Wicklow et al., 2014; Wu et al., 2013).”

While we acknowledge the potential for using maternal-only KO controls to underestimate differences between control and KO samples, we believe this approach does not introduce false positives in our RNA-seq and ATAC-seq experiments, only the possibility of more conservative conclusions. This minimizes the risk of overestimating the molecular impact.

We appreciate the reviewer’s recommendation regarding the use of overly assertive terms. Upon careful review of the manuscript and response letter, we could not find instances of the term “absolutely priority.” However, we do use the term “pivotal” and would prefer to retain it as we believe it accurately reflects the importance of the findings presented in our manuscript.

Thank you for your thoughtful comments and suggestions! We hope this response clarifies our rationale and addresses the concerns.

---

The following is the authors’ response to the original reviews.

**Public Reviews:**

**Reviewer #1 (Public review)**
Summary:Numerous mechanism and structural studies reported the cooperative role of Oct4 and Sox2 during the establishment of pluripotency during reprogramming. Due to the difficulty in sample collection and RNA-seq with low-number cells, the precise mechanisms remain in early embryos. This manuscript reported the role of OCT4 and SOX2 in mouse early embryos using knockout models with low-input ATAC-seq and RNA-seq. Compared to the control, chromatin accessibility and transcriptome were affected when Oct4 and Sox2 were deleted in early ICM. Specifically, decreased ATAC-seq peaks showed enrichment of Motifs of TF such as OCT, SOX, and OCT-SOX, indicating their importance during early development. Moreover, by deep analysis of ATAC-seq and RNA-seq data, they found Oct4 and Sox2 target enhancer to activate their downstream genes. In addition, they also uncovered the role of OS during development from the morula to ICM, which provided the scientific community with a more comprehensive understanding.Strengths:On the whole, the manuscript is innovative, and the conclusions of this paper are mostly well supported by data, however, there are some issues that need to be addressed.Weaknesses:Major Points:(1) In Figure 1, a more detailed description of the knockout strategy should be provided to clarify itself. The knockout strategy in Fig1 is somewhat obscure, such as how is OCT4 inactivated in Oct4mKO2 heterozygotes. As shown in Figure 1, the exon of OCT4 is not deleted, and its promoter is not destroyed. Therefore, how does OCT4 inactivate to form heterozygotes?

Thank you for helping clarify this. We will add a detailed description of the knockout strategy in the legends for Figure 1A and 1B, as shown below. Note that the same strategy was used by Nichols *et al* (Cell, 1998).

Figure 1A. Schemes of mKO2-labeled *Oct4* KO (*Oct4^mKO2^*) and *Oct4^flox^* alleles. In the *Oct4^mKO2^* allele, a PGK-pac∆tk-P2A-mKO2-pA cassette was inserted 3.6 kb upstream of the *Oct4* transcription start site (TSS) and a promoter-less FRT-SA-IRES-hph-P2A-Venus-pA cassette was inserted into Oct4 intron 1. The inclusion of a stop codon followed by three sets of polyadenylation signal sequences (pA) after the *Venus* cassette ensures both transcriptional and translational termination, effectively blocking the expression of Oct4 exons 2–5.

Figure 1B. Schemes of EGFP-labeled *Sox2* KO (*Sox2^EGFP^*) and *Sox2 ^flox^* alleles. In the *Sox2 Sox2^EGFP^* allele, the 5’ untranslated region (UTR), coding sequence and a portion of the 3’ UTR of *Sox2* were deleted and replaced with a PGK-EGFP-pA cassette. Notably, 1,023 bp of the *Sox2* 3’UTR remain intact.

(2) Is ZP3-Cre expressed in the zygotes? Is there any residual protein?

This is indeed a very important issue. Here is why we think we are on the safe side. ZP3 is specifically expressed in growing oocytes, thus making ZP3-Cre a widely used tool for deleting maternally inherited alleles. When we crossed *Oct4^flox/flox^*; _ZP3-Cre^-^_females with *Oct4^flox/flox^*; *ZP3-Cre^+^* males, we got *ZP3-Cre^+^ Oct4^flox/flox^* but no Oct4*flox/∆* or Oct4*∆/∆* pups, suggesting that the paternally inherited *ZP3-Cre* allele is not functionally active in zygotes, which is consistent with reports from other researchers (e.g. Frum, *et al.,* Dev Cell 2013; Wu, *et al.,* Nat Cell Biol 2013).

(3) What motifs are enriched in the rising ATAC-seq peaks after knocking out of OCT4 and SOX2?

The enriched motifs in the rising ATAC-seq peaking in *Oct4* KO and *Sox2* KO ICMs are the GATA, TEAD, EOMES and KLF motifs, as shown in Figure 4A and Figure supplement 7.

(4) The ordinate of Fig4c is lost.

Thank you for pointing this out. The y-axis is average normalized signals (reads per million-normalized pileup signals). We will add it in the revised version.

(5) Signals of H3K4me1, H3K27ac, and so on are usually used to define enhancers, and the loci of enhancers vary greatly in different cells. In the manuscript, the authors defined ATAC-seq peaks far from the TSS as enhancers. The definition in this manuscript is not strictly an enhancer.

Thank you for this insightful comment. We analyzed the published H3K27ac ChIP-seq data of mouse ICM at 94-96 h post hCG (B. Liu, et al., Nat Cell Biol 2024) to assess the enrichment of H3K27ac around our ATAC-seq peaks. Unfortunately, the data quality is poor, e.g., inconsistent across replicates (Author response image 1A), and shows little enrichment around the well-defined enhancers (Author response image 1B). Nevertheless, as we admit not all the distal ATAC-seq peaks or open chromatin regions are enhancers, we have replaced “enhancers” with “open chromatin regions”, “ATAC-seq peaks” or “putative enhancers”.

**Author response image 1. sa3fig1:** Analysis of the published H3K27ac ChIP-seq dataset of mouse ICM at 94-96 h post hCG (B) (Liu, et al., Nat Cell Biol 2024). A. ChIP-seq profiles of H3K27ac over the decreased, unchanged and increased ATAC-seq peaks in our Oct4-KO late ICMs. To exclude spurious peaks, only strong unchanged peaks (57,512 out of 142,096) were used in the analysis. B. IGV tracks displaying ATAC-seq and H3K27ac ChIP-seq profiles around *Dppa3* and *Oct4*. Red boxes mark the known OCT-SOX enhancers.

(6) If Oct4 and Sox2 truly activate sap 30 and Uhrf 1, what effect does interfere with both genes have on gene expression and chromatin accessibility?

This is indeed an interesting question. Unfortunately, we have not conducted this specific experiment, so we do not have direct results. However, Sap30 is a key component of the mSin3A corepressor complex, while Uhrf1 regulates the establishment and maintenance of DNA methylation. Both proteins are known to function as repressors. Therefore, we hypothesize that interfering with these two genes could alleviate repression of some genes, such as trophectoderm markers, similar to what we have observed in Oct4 KO and Sox2 KO ICMs.

**Reviewer #2 (Public review):**
In this manuscript, Hou et al. investigate the interplay between OCT4 and SOX2 in driving the pluripotent state during early embryonic lineage development. Using knockout (KO) embryos, the authors specifically analyze the transcriptome and chromatin state within the ICM-to-EPI developmental trajectory. They emphasize the critical role of OCT4 and the supportive function of SOX2, along with other factors, in promoting embryonic fate. Although the paper presents high-quality data, several key claims are not well-supported, and direct evidence is generally lacking.Major Points:(1) Although the authors claim that both maternal KO and maternal KO/zygotic hetero KO mice develop normally, the molecular changes in these groups appear overestimated. A wildtype control is recommended for a more robust comparison. (a complementary comment from the reviewer: “Both maternal KO and maternal-zygotic KO in this study exhibited phenotypic consistency but molecular disparity. Specifically, both KO and control groups could develop normally; however, their chromatin landscapes and transcriptomic profiles showed different. This raises the question of whether the molecular differences are real. We suggest that inclusion of a completely wild-type control group would make the comparison more robust.”)

Thank you for your feedback as this point was obviously not clear in the manuscript. Here is our explanation: Mouse embryos with a maternal KO or zygotic heterozygous KO of Oct4 or Sox2 show no noticeable phenotype or molecular difference (Figure 2-figure supplement 4A) (Avilion et al., 2003; Frum et al., 2013; Kehler et al, 2004; Nichols et al., 1998; Wicklow et al., 2014; Wu et al., 2013). We have clarified this point in the revised manuscript.

(2) The authors assert that OCT4 and SOX2 activate the pluripotent network via the OCT-SOX enhancer. However, the definition of this enhancer is based solely on proximity to TSSs, which is a rough approximation. Canonical enhancers are typically located in intronic and intergenic regions and marked by H3K4me1 or H3K27ac. Re-analyzing enhancer regions with these standards could be beneficial. Additionally, the definitions of "close to" or "near" in lines 183-184 are unclear and not defined in the legends or methods.

Thank you for this insightful and helpful comment. As stated in the response to Reviewer #1’s point (5), we have replaced “enhancers” with “open chromatin regions”, “ATAC-seq peaks” or “putative enhancers”.

The definition of "close to" or "near" in lines 183-184 is in the legend of Figure 2E and Methods. In the GSEA analysis, Ensembl protein-coding genes with TSSs located within 10 kb of ATAC-seq peak centers were included, so that some of the intronic ATAC-seq peaks were taken into consideration. We have also added the information in the main text of the revised manuscript.

(3) There is no evidence that the decreased peaks/enhancers could be the direct targets of Oct4 and Sox2 throughout this manuscript. Figures 2 and 4 show only minimal peak annotations related to OCT and SOX motifs, and there is a lack of chromatin IP data. Therefore, claims about direct targets are not substantiated and should be appropriately revised.

Yes indeed, you have a point. In Figure Supplement 3C, we analyzed the published Sox2 CUT&RUN data from E4.5 ICMs (Li et al., Science, 2023), which demonstrates that the reduced ATAC-seq peaks in our Sox2 KO ICMs are enriched with the Sox2 CUT&RUN signals. Unfortunately, we did not to find similar published data for Oct4 in embryos. We have removed the statement indicating that these are the direct targets in the revised manuscript.

(4) Lines 143-146 lack direct data to support the claim. Actually, the main difference in cluster 1, 11 and 3, 8, 14 is whether the peak contains OCT-SOX motif. However, the reviewer cannot get any information of peaks activated by OCT4 rather than SOX2 in cluster 1, 11.

Thank you for the comment that we hope we can clarify.

Lines 143-146 are: “Notably, the peaks activated by Oct4 but not by Sox2 in the ICM tended to be already open at the morula stage (Figure 2B, clusters 1 and 11), whereas those dependent on both Oct4 and Sox2 became open in the ICM (Figure 2B, clusters 3, 8 and 14).”

We agree with you that clusters 3/8/14 are more enriched in OCT-SOX motifs than clusters 1/11. However, this is consistent with our observation that accessibility of peaks in clusters 1 and 11 relies mainly on Oct4, while accessibility in clusters 3, 8, 14 depends on both Oct4 and Sox2. But maybe the term “activate” is misleading. We have rephrased the text as below:

“Notably, compared to the peaks that depend on Oct4 but not Sox2 (Figure 2B, clusters 1 and 11), those reliant on both Oct4 and Sox2 show greater enrichment of the OCT-SOX motif (Figure 2B, clusters 3, 8 and 14). The former group was generally already open in the morula, while the latter group only became open in the ICM. “

Minor Points:(1) Lines 153-159: The figure panel does not show obvious enrichment of SOX2 signals or significant differences in H3K27ac signals across clusters, thus not supporting the claim.

We hope to be able to explain this.

Line 153-159 refer to two datasets: Figure Supplement 3C and 3D.

In Figure Supplement 3C, the average plots above the heatmaps show that the decreased ATAC-seq peaks (the indigo lines) have higher enrichment with Sox2 CUT&RUN signals than the increased or unchanged peaks (the yellow and light blue lines, respectively).

In Figure Supplement 3D, the average plots indicate that H3K27ac signals around the center of the decreased ATAC-seq peaks (the indigo line) show higher enrichment compared to the unaltered and decreased groups (the light blue and yellow lines, respectively). Notably, H3K27ac enrichment appears slightly offset from the central nucleosome-free regions.

(2) Lines 189-190: The term "identify" is overstated for the integrative analysis of RNA-seq and ATAC-seq, which typically helps infer TF targets rather than definitively identifying them.

You are right. We have replaced “identify” with “infer” in the revised manuscript.

(3) The Discussion is lengthy and should be condensed.

We have shortened the discussion in the revised manuscript.